# UniPoint-LLM: Unifying Point Cloud Understanding and More Controllable 3D Generation

## Abstract

We introduce UniPoint-LLM, which integrates point cloud understanding ability into Image Multimodal Large Language Model(MLLM) and enable more flexible and controllable natural language-driven 3D generation, realized the unified process of point cloud understanding and generation. Unlike traditional text-to-3D methods with limited prompt inputs or constrained parameters, UniPoint-LLM allows users to input natural language description to specify their requirements. By aligning image and point cloud modalities through joint training and weights sharing, UniPoint-LLM also achieves two modalities' understanding. Experiments demonstrate that UniPoint-LLM offers users greater flexibility and control in generating desired 3D objects and the effectiveness of our Multimodal Universal Token Space(MUTS) in understanding both images and point clouds. These experiments validates its potential value in practical applications of 3D generation and interactive design.

## 1 Introduction

In recent years, multimodal large language models(MLLMs) have made significant breakthroughs in the fields of natural language processing and computer vision. Many works have demonstrated the ability to bridge the gap between natural language and multiple modalities, including images, audio, and videos, providing powerful capabilities for machine intelligence.(Alayrac et al., 2022; Li et al., 2023; Liu et al., 2023; Ye et al., 2023; Zhu et al., 2023; Su et al., 2023) However, there has been relatively less research and application of multimodal large language models in handling 3D data, particularly in understanding and generating point cloud at the same time.

The complexity and unordered nature of point cloud data present challenges in understanding and generating. Traditional approaches often separate the understanding of point clouds from the 3D generation process. For text-to-3D generation, current methods rely on short prompts for control (Lin et al., 2023; Jun & Nichol, 2023). In terms of point cloud understanding, some works map point clouds to a 2D space for learning and understanding (Zhang et al., 2022), while others (Xu et al., 2023) directly align a single point cloud modality to the LLM space. However, these methods often have limited capabilities of generation according to detailed requirements or multimodality understanding, which restrict the flexibility of expressing to generate 3D objects and multimodality interaction method by user preferences. Recently, although some work (Guo et al., 2023) has aligned point cloud to a modality space, this alignment method requires more complex data association pairs. When they adding a new modality, it requires building the new modality data pairs that align with all previous modalities.

Our research takes inspiration from the latest model, DALL-E3 (202, 2023). It is an innovative 2D generation model that combines chatGPT with image generation, enabling fine-grained control over the content of generated images. Although DALL-E3 has limitations in improving generation quality, it significantly enhances the controllability and freedom for users during the generation process. This led us to ask a question: **whether we can propose a unified simple model to fulfill the desired requirements for understanding point clouds even more vision modalities and achieving more detailed and controllable text-to-3D generation using natural language?**

Therefore, this paper introduces UniPoint-LLM, a unified framework for image and point cloud understanding and more flexible text-to-3D generation. UniPoint-LLM enables simultaneous understanding of both image and point cloud modalities because image also holds a very important position in the works of point clouds. Drawing inspiration from DALL-E3 (202, 2023), our approach leverages the advantages of large language models in terms of controllability and freedom. By embedding text into the large language model, we achieve a more flexible and controllable text-to-3D generation process.

Our contributions are as follows:

- **Text-to-3D generation based on MLLM.** We are the first to introduce 3D point cloud flexible generation capabilities into Multimodal Large Language Models(MLLMs) without sacrificing their language understanding abilities. Unlike previous methods (Guo et al., 2023), which relied on flows, our approach is based on 3D diffusion generation decoders which leads to greater diversity in our generated outputs.

- **Mapping multimodalities to a unified space.** We propose a simple Multimodal Unified Token Space (MUTS) to handle the mapping of image and point cloud data to a unified space. This eliminates the need for additional alignment operations for different modalities in large language models. Through LLM's inherent understanding, LLMs can handle both image and point cloud modalities. This approach also facilitates future extensions to incorporate additional multimodality.

## 2 RELATED WORK

This section will introduce the research work related to two aspects of 3D generation and multimodal learning as discussed in this paper.

**3D Generation.** 3D generation is a task aimed at creating realistic and diverse 3D models from different inputs (such as text, images, sketches, or point clouds). This task is challenging and requires a deep understanding of the shape, structure, texture, and semantics of 3D objects. The main methods currently include parametric methods and non-parametric methods. Parametric methods use predefined templates or primitives to represent 3D shapes, such as voxels, meshes, point clouds, or implicit functions.(Smirnov et al., 2020; Palafox et al., 2021) These methods can generate high-resolution and high-fidelity smooth continuous 3D models. However, these methods also have some limitations, such as high computational cost, fixed topology, or difficulty in handling complex geometry. Non-parametric methods use generative models to learn the distribution of 3D shapes from data, such as Generative Adversarial Networks (GANs)(Paper, 2021), Variational Autoencoders (VAEs)(Kingma & Welling, 2013), or Normalizing Flows (Yang et al., 2019). These methods can generate diverse and novel 3D models without relying on predefined templates or primitives. However, these methods also face some challenges, such as mode collapse, decoupling, or evaluation issues. Recently, with the rise and rapid development of diffusion models in the 2D field, more and more 3D generation research has begun to adopt diffusion models. Shap-e (Jun & Nichol, 2023) is a Transformer-based 3D generation model proposed by OpenAI that supports text-to-3D conversions. It can generate realistic and diverse 3D models based on natural language descriptions. In our research, we use it as decoder to complete the generative network. The original Shap-e model uses CLIP (Radford et al., 2021) to embed text, and CLIP is trained on billions of text-image pairs. Therefore, when performing text-to-3D conversion, users can usually only generate the 3D objects they expect by adjusting some parameters and providing relatively shot text prompts, and cannot achieve more detailed control.

**Multimodal Alignment.** Multimodal alignment aims to find the correspondence between different data modalities, such as text, images, videos, audio, and 3D point clouds. For large language models (LLMs), understanding visual features is very important. There are two types of methods: multi-space methods and single-space methods. Multi-space methods map different modalities to their own spaces and interact and transform through mechanisms such as attention, gating, contrastive learning, or latent variables. BLIP2 (Li et al., 2023) proposed QFormer, which uses a multi-branch structure to encode text and images separately, and then aligns them through cross-modal attention and contrastive learning. LLaVA (Liu et al., 2023) uses a single-branch structure to jointly encode text and images, and then generates descriptions through latent variables. PandaGPT (Su et al.,

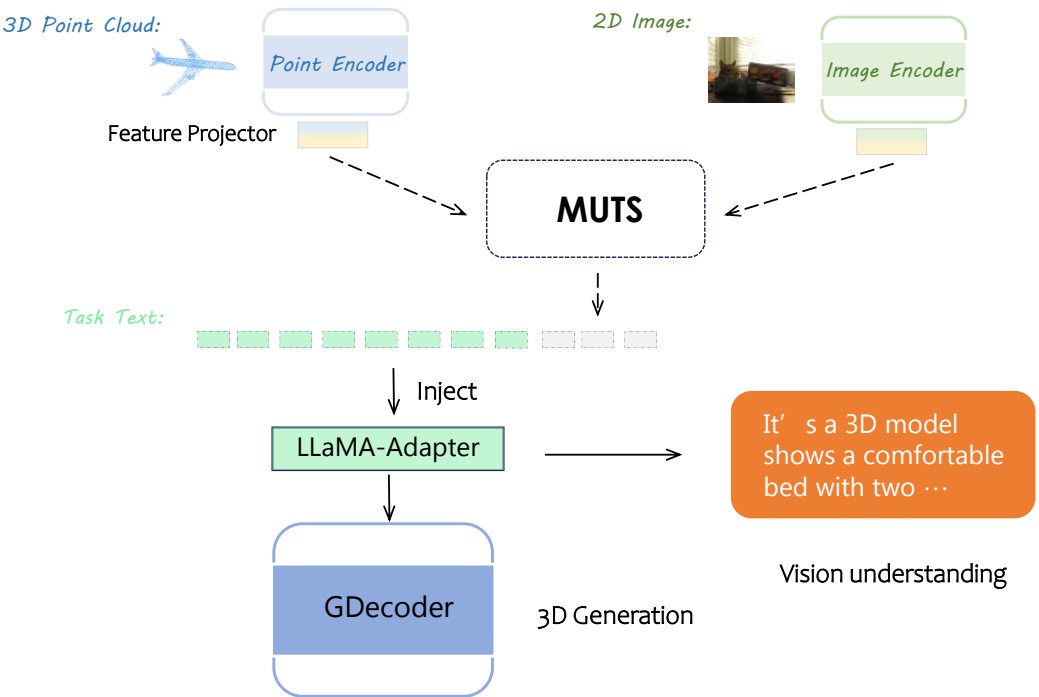

Figure 1: An overview of Uni-LLM. It extracts preliminary features of different modalities through different modality encoders, then obtains global features through MUTS, after that, the global features align to the text space, and then injects into LLaMA-Adapter. We extract the last hidden layer of llama-adapter as text condition to guide the generation task, and the text output corresponding understanding.

2023) also uses a single-branch structure to jointly encode text and 3D point clouds, and then generates text or point clouds through autoregressive decoding. These methods can retain the information of the original data, but it is difficult to extend to new modalities. Single-space methods align different modalities into a single space. ImageBind (Girdhar et al., 2023) uses a Transformer-based model to align images, audio, text, and other modalities in the image space, and generates images or text through autoregressive decoding. PointBind (Guo et al., 2023) also establishes data pairs between point clouds and other modalities, and performs aligned contrastive training. These methods can achieve cross-modal alignment, but they require more data association pairs, and adding new modalities requires alignment with all previous modalities. Our method MUTS maps different modalities(image and point cloud) to a single space, and only needs additional data pairs that align the new modality with the text.

## 3 METHODS

The overall architecture of the model is shown in Figure 1. In Section 3.1, we explain in detail the Multimodal Universal Token Space (MUTS), and in Section 3.2 we introduce our model's text-to-3D method. In Section 3.3, we delve into the detail of training data.

### 3.1 MULTIMODAL UNIVERSAL TOKEN SPACE

The Multimodal Universal Token Space (MUTS) is a model for handling multimodal inputs (images and point clouds). It consists of independent encoders for each modality, a mapping alignment layer, and a global feature extraction block.

In MUTS, for each modality input, it is first encoded through the corresponding pre-trained encoder. Then, the image features and point cloud features are mapped to the token space with the same

dimensions through the mapping alignment layer. The mapping alignment layer is a simple and effective method that can map feature vectors of any dimension to token vectors of fixed dimensions.

We represent it as the following formulas:

$$F_{img} = Map_{img}(IEncoder(I)) \tag{1}$$

$$F_{pc} = Map_{pc}(PCEncoder(P)) \tag{2}$$

where $I$ and $P$ represent image and point cloud inputs respectively. $IEncoder$ and $PCEncoder$ represent image and point cloud encoders, $Map_{img}$ and $Map_{pc}$ represent image and point cloud mapping alignment layers.

To align the mapped tokens with the text space and pass them to the LLM, we concatenate the image or point cloud features with a global feature which is initialized randomly at the beginning, then use a pre-trained Transformer Block for global feature extraction. After feature extraction, we use the global feature as the final visual feature representation, as shown in Figure 2.

The overall formula is as follows:

$$F_{muts} = \begin{bmatrix} F_{global} & F_{txt} \end{bmatrix} \tag{3}$$

where the global feature vector is represented as $F_{global}$, and the text feature vector is represented as $F_{txt}$. Then we connect them together to form the final multimodal universal token space input $F_{muts}$, finally passing $F_{muts}$ as input to LLM to gradually learn visual representation information, and generate corresponding output after fusing visual features.

In summary, Multimodal Universal Token Space (MUTS) is a module for handling multimodal inputs such as images and point clouds. Using MUTS for visual feature extraction can map content of different modalities into the same space distribution. This unified representation helps to extend new modalities later. Users only need to prepare a pre-trained encoder for the new modality and corresponding data text pairs to easily integrate new modalities into the system. This modular design makes MUTS highly flexible and scalable in dealing with different types of multimodal inputs.

**Settings:** To implement MUTS, we used llama-adapter (Gao et al., 2023) as the base model, which is a multimodal language model pre-trained on text and image. Llama-adapter uses seperate softmax and zero initial gate to gradually learn visual feature knowledge while not disrupting existing knowledge structures when introducing new knowledge. By using adapter modules to convert image features into text features, llama-adapter has already aligned image space with text space. On this basis, we expanded the point cloud modality and shared the MUTS weights with the image modality.

For point cloud input, we used EPCL (Huang et al., 2022) as encoder, which is a point cloud processing model based on CLIP that can extract global and local features from unordered sparse point clouds. EPCL is an efficient and effective point cloud learner that can directly train high-quality point cloud models with frozen pruning models. We selected a EPCL detection model trained on ScanNet (Dai et al., 2017) dataset.

**Training Details:** Our MUTS method adopts a two-stage training strategy. In the first stage, we freeze the weights of MUTS network and overall llama-adapter, only training the point cloud encoder. The purpose of this is to make the feature distribution extracted by the point cloud encoder consistent with image features in shared universal space. In this stage, we do not train on entire network in order to avoid interference from other components in overall network on point cloud encoder.

In second stage, we active the entire network for joint training. This means that we train point cloud encoder, image encoder, llama-adapter at same time to further adjust parameters of model so that it performs better in multimodal tasks. Through this two-stage training method, we can fully utilize pre-trained llama-adapter model and better handle multimodal information.

## 3.2 TEXT-TO-3D GENERATION USING LLM

We propose an improved text-to-3D generation method based on LLM text features. Compared with existing text-to-3D generation methods that usually use CLIP text encoders to process text prompts, we choose LLM with a larger text feature space as the basis. By using LLM as a text encoder, our

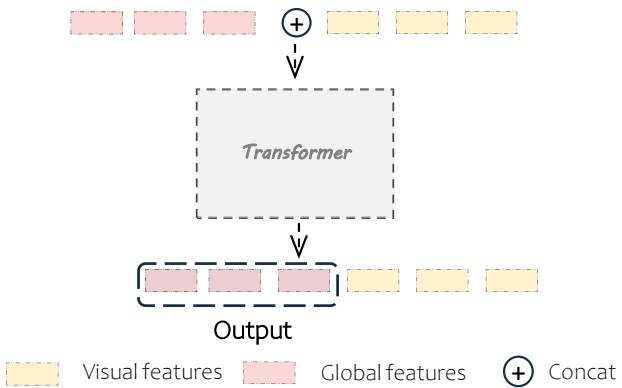

Figure 2: Muts process.

generative model can better handle text prompts or descriptions to generate high-quality 3D objects. In Figure 3, examples of models generated quality by UniLLM and Shap-e are shown (short prompt), which demonstrate that in prompt cases, we can get the same or better quality of generation.

**Training Details:** In our method, we use the output of the last hidden space of LLM as the text condition for the generative decoder. In order to obtain an LLM model that can effectively extract text features, we first carry out the first stage of training. In this stage, we train the LLM model by multimodal data pairs to ensure fully understanding of the input text information. Then, we freeze the LLM model and only use it as an encoder for the text, without any further parameter updates.

Next, we use the cross attention to extract learnable text features to align with the text input dimension of shap-e. At the same time, we can capture important text features related to 3D generation in the output of LLM. Subsequently, we and align the text feature into the shap-e decoder. In this way, the generation process is constrained by LLM text features, thus producing a 3D model consistent with the input text.

Through the above training process, we can fully utilize the text feature expression ability of the LLM model and combine it with the shap-e decoder to achieve more accurate and controllable text-to-3D generation process.

### 3.3 TRAINING DATA

We use the Cap3d dataset (Luo et al., 2023) as the base dataset, which extracts some point cloud objects from Objaverse (Deitke et al., 2023) and provides detailed descriptions of 3D objects using BLIP2 (Li et al., 2023) and GPT (cha, 2023), as well as multi-view information after rendering with Blender. This dataset includes 660k object point clouds and corresponding caption text. Based on this dataset, we can train our model to make UniPoint-LLM have point cloud understanding capabilities.

**3D Q&A:** In order to achieve the model's 3D Q&A and reasoning capabilities, we use the Instruct tuning training method. In terms of data generation, we also use the Cap3d dataset and call GPT to generate 5-10 normal Q&A and reasoning-related Q&A pairs for each caption and object to simulate interactive Q&A with users.

**Natural Language Prompts:** Although the Cap3d dataset provides diverse text descriptions that contain some detailed information, its length is only about 10 words, which is still a short caption, not much different from a prompt, and cannot match the real user's language. In order to better simulate user's natural language input and provide more detailed descriptions of the generated 3D models, we use GPT to expand the original text caption. We designed several data examples, allowing GPT to imitate and expand based on the original caption, to simulate user's natural speaking style and more detailed description of objects.

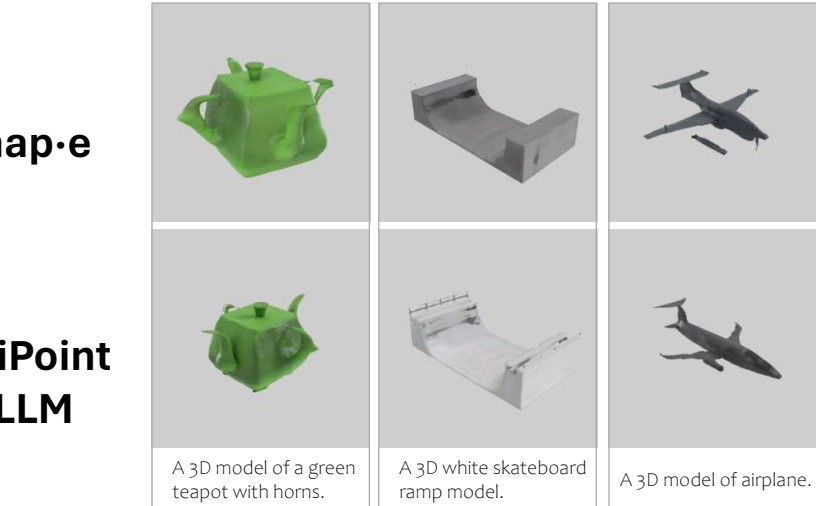

**Shap·e**

**UniPoint -LLM**

A 3D model of a green teapot with horns. | A 3D white skateboard ramp model. | A 3D model of airplane.

Figure 3: The prompt-based generation.

## 4 EXPERIMENTS

To evaluate the performance of our method in 3D generation, we conducted a series of experiments on the Cap3d dataset.

### 4.1 EXPERIMENTAL SETTINGS

Firstly, we divided the dataset into training and testing samples following the original division of Cap3d. When testing the generation model metrics, we selected a test sample size of 2000. In addition, we used Blender to render the generated objects from 8 angles to obtain multi-view information.

Secondly, to compare the performance of our model with shap-e in terms of long text descriptions, we also fine-tuned the models using description data.

Through these experimental settings, we can accurately evaluate and analyze the performance of our method in 3D generation tasks.

### 4.2 EVALUATION METRICS:

Following Cap3d, we further tested the generation effect of UniPoint-LLM on the dataset's prompt and evaluated it using the following metrics:

**FID:** (Fréchet Inception Distance) FID is a commonly used metric to measure the distribution similarity between generated images or point clouds and real data. A lower FID value indicates that the generated results are more similar and realistic to the real data.

**CLIP Correlation Metrics:** CLIP correlation metrics use Blender to render real 3D objects into images, then use CLIP to encode these images, and calculate their similarity with the original prompt. These metrics help us evaluate the consistency and correlation between the generated results and the prompts.

### 4.3 RESULTS

**Prompt-to-3D:** We compared our method with fine-tuned point-e and shap-e on Cap3d, Table 1. shows the test results using Cap3d generated 3D objects. On the FID metric, our model usually outperforms the same finetuned base model. For CLIP-related metrics, the results show that UniPoint-

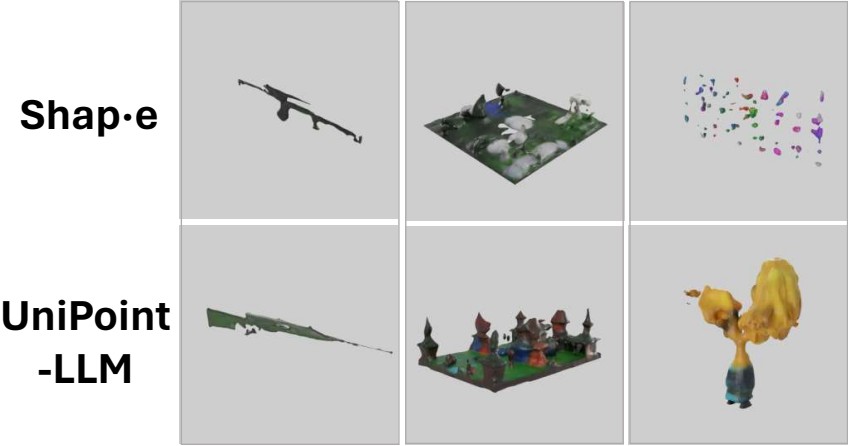

Figure 4: The description-based generation.

LLM does not significantly improve the generation effect compared with directly finetuning the original model, and even scores slightly lower on CLIP R@1. This may be due to the distribution difference between Uni-LLM and CLIP's text embedding space, there is still a slight difference in the distribution similarity of the final generated real data. But it can be proved that our model still maintains a high generation quality when using short prompts.

Table 1: **Text-to-3D on Cap3D.** Point-llm compared with the generation models after fine-tuning on Cap3D.

|  | FID ↓ | ClIP Score | CLIP R@1 | R-precision(2k) | |
|---|---|---|---|---|---|
|  |  |  |  | R@5 | R@10 |
| Ground Truth | - | 81.6 | 32.7 | 55.1 | 64.3 |
| Point-E | 32.8 | 75.6 | 12.4 | 28.1 | 36.9 |
| Shape-E(NeRF) | 48.2 | 78.1 | 18.3 | 35.1 | 43.5 |
| Shape-E(STF) | 35.5 | 79.1 | 20 | 38.8 | 47.3 |
| Uni-LLM(NeRF) | **39.4** | **78.2** | **17.4** | **35.8** | **44.2** |
| Uni-LLM(STF) | **34.7** | **78.8** | **19.6** | **39.4** | **47.5** |

**Description-to-3D:** In order to evaluate the model's generation ability closer to human descriptive language, we test some description samples on UniPoint-LLM and Shap-e. As shown in Figure 3, in order to compare the differences between models in more detail, we divided the text description categories into three parts: **description of specific objects**, **description of scenes**, and **description of abstract things**. For these three types of descriptions, we selected one sample for display respectively, from which it can be clearly seen that our model has higher accuracy. The description are:1.Can you please generate a rifle that is mostly green, with some white at the muzzle? It should have a long barrel, a stock, and a scope attached to the top. The green color should be a vibrant, military-style green, while the white muzzle can be a more subdued, matte finish. 2.Please create a 3D model of a small town. The town should have many houses, trees, and rivers.3.Imagine a figure that makes you feel frightened and uneasy. What does it look like?"

- **Description of Objects:** When UniPoint-LLM describes objects, it shows better generation effects in terms of aspects like the trigger, sight, body length, and color. In contrast, the original shap-e might only capture simple information, such as "gun" and "green", and due to the length of the text, it might generate relatively abstract effects that cannot handle some detailed requirements well.

- **Description of Scenes:** From the description of scenes, it can be seen that our model includes elements such as houses and trees in the generated description, while shap-e ignores these elements and only generates a relatively simple grassland scene with a river. This fully demonstrates that when generating scenes with more elements, our model has better controllability in handling elements in the scene and can extract the content of elements in the text well.

- **Description of Abstract Things:** Interestingly, we also compared the descriptions of two models on abstract things. We asked them to describe an object containing scary elements separately. In this example, shap-e failed directly, while UniPoint-LLM generated an abstract image of a two-headed humanoid. This indicates that UniPoint-LLM may imply more high-level semantics in the process of generating text, thus proving its strong controllability in the generation model.

## 5 ABLATION STUDY

To evaluate the effectiveness of the MUTS, we conducted an ablation study to explore the effectiveness of the fusion process of image and point cloud modalities, and use the accuracy of 3D object classification on ModelNet40 (Vishwanath et al., 2009) as an evaluation metric. The specific results are shown in Table 2. **Zero-shot 3D object classification.** The task of 3D object classification is to prompt the model to generate the object type based on the given point cloud. This tests the LLM's understanding ability using prior knowledge. We use the test split of the ModelNet40 (ModelNet) dataset to test our model's understanding of 3D objects. ModelNet40 contains point clouds of 40 different object categories. We initially considered completing the classification task by multiple choice, letting our model choose a category to represent the input point cloud. However, we consider the length of the ModelNet40 overall category text and the realistic environment and user dialogue situations, we chose to directly generate descriptive words or phrases related to the point cloud as output, but as a result we could not directly parse its response for evaluation. Therefore, we input our descriptive sentences and categories into GPT at the same time, and use the accuracy of GPT model's selection of the ModelNet40 categories as the accuracy of our model. We train two exper-

Table 3: **Accuracy of MUTS and Hard Alignment on ModelNet40.** The results show the classification accuracy for the two aligment methods.

| Alignment Methods | Acc(%) |
|---|---|
| Hard Alignment | 30.94 |
| MUTS | **32.28** |

imental models, one using the MUTS module to align the point cloud modality with the text space, and the other directly aligning the point cloud modality to the text space. The results show that when using the MUTS module to simultaneously fuse image and text modalities, the understanding ability of a single modality (point cloud) does not decrease, which proves the effectiveness of our module.

## 6 CONCLUSION

In this paper, we propose UniPoint-LLM, a framework that combines the understanding of large language models with images and point clouds and achievs a more flexible and controllable natural language-driven 3D generation. We emphasize the importance of multimodal understanding (especially images and point clouds) in the 3D field and MLLM. By introducing a Multimodal Unified Feature Space (MUTS), we achieve multimodal understanding of images and point clouds. At the same time, our method uses the text features of large language models for 3D generation. Through experiments, our method can not only generate high-quality 3D objects from short text prompts, but also allow users to generate more accurate and controllable objects through more detailed text descriptions. This provides a beneficial exploration direction for the research and application of text-to-3D generation. We believe that combining the controllability and freedom of large language models can further promote the perception, interaction, and generation effects of intelligent systems in three-dimensional scenes, and promote the development and application of artificial intelligence technology.

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
