# OpenReview forum: "PointMLLM: Aligning multi-modality with LLM for point cloud understanding, generation and editing"
_ICLR.cc/2024/Conference — ICLR 2024 Conference Withdrawn Submission_

### Official Review · Reviewer_UrQG · 2023-10-30

**Soundness:** 3 good
**Presentation:** 4 excellent
**Contribution:** 3 good
**Rating:** 6
**Confidence:** 5

**Summary:**

In this paper, the authors introduce UniPoint-LLM, a framework that combines point cloud and image features with multimodal large language models (MLLMs) to achieve vision understanding and 3D generation. The framework utilizes a Multimodal Universal Token Space (MUTS) to handle multimodal inputs and maps them to a unified space. This is the first paper to complete text-to-3D generation using MLLMs and to map multimodalities to a unified space.

**Strengths:**

The paper has several notable strengths:
1. The paper is well-written and easy to understand, with a clear and effective approach.
2. This paper is the first attempt to generate text-conditioned 3D using MLLMs, providing a new perspective on text-to-3D generation.

**Weaknesses:**

1. Unclear Motivation: The paper posits that UniPoint-LLM is distinct from traditional text-to-3D methods due to its ability to handle "unlimited prompt inputs or unconstrained parameters." However, existing methods such as TextMesh [1] and DreamTime [2] have demonstrated capabilities in processing long and flexible descriptions. This raises questions regarding the necessity of introducing MLLM for handling unlimited texts.

2. Lack of Experiments: The authors assert that UniPoint-LLM is advantageous for vision understanding and 3D generation. Nonetheless, the section on vision understanding lacks quantitative results, particularly a comparative analysis with the state-of-the-art method JM3D [3] on benchmarks like ModelNet40 and ScanObjectNN. Additionally, the qualitative results presented for 3D generation are sparse. The experiments, as they stand, fall short of substantiating the claimed contributions.

3. Novelty: The proposed adjustment of substituting the text encoder in established methods like PointE [4] with MLLM appears to be incremental, and does not evidently constitute a novel contribution.

[1] TextMesh: Generation of Realistic 3D Meshes From Text Prompts, arXiv:2304.12439, 2023

[2] Dreamtime: An Improved Optimization Strategy for Text-to-3D Content Creation, arXiv:2306.12422

[3] Beyond First Impressions: Integrating Joint Multi-modal Cues for Comprehensive 3D Representation, ACM MM23

[4] Point-E: A system for generating 3D point clouds from complex prompts, arXiv:2212.08751

**Questions:**

1. A more robust set of quantitative results could indeed bolster the claims made in the paper. It's unfortunate that there isn't a more extensive experimental evaluation, akin to what's presented in works like JM3D [1] or ULIP [2]. Could you provide more detailed results to substantiate your claims?

2. I am curious about whether the integration of MLLM introduces additional computational overhead. Are there any quantitative metrics available to gauge the trade-off involved?

3. It would be beneficial to elucidate the distinctions between UniPoint-LLM and other LLM-based 3D models such as JM3D-LLM [3] and Point-LLM [4]. At a glance, there seems to be substantial overlap among these works. Could you shed light on the unique contributions of UniPoint-LLM in this context?

4. It's noted that the foundation of MUTS is a pretrained transformer block. Could you specify the dataset on which this block was trained? Furthermore, do you believe that employing this pretrained block is a judicious choice for subsequent training, especially given the mixed data of images and point clouds involved?

[1] ULIP: Learning a unified representation of language, images, and point clouds for 3D understanding, CVPR 23.
[2] JM3D & JM3D-LLM: Elevating 3D Representation with Joint Multi-modal Cues, arXiv:2310.09503.
[3] PointLLM: Empowering large language models to understand point clouds, arXiv:2308.16911.

---

### Official Review · Reviewer_JaY5 · 2023-10-31

**Soundness:** 2 fair
**Presentation:** 2 fair
**Contribution:** 2 fair
**Rating:** 3
**Confidence:** 3

**Summary:**

This paper proposes a Multimodal Universal Token Space (MUTS) for joint image and point cloud learning. The proposed method leverages large language model and align the latent space of image and point cloud. The paper shows an improved performance on text-to-point cloud generation. However, the overall writing of the paper is poor and technical details are missing. The overall model and training details are not clear. Besides, it’s also strange to perform experiments on 3D point cloud generation to evaluate point cloud understanding. Overall the paper provides some insights to the community, however, the technical contribution and the experimental design are weak.

**Strengths:**

1. The method is straight-forward and simple to implement.
2. The method shows better generation quality than Shap-E.

**Weaknesses:**

1. The overall writing and structure are poor. Although the method is simple, the authors do not provide clear motivation for method design. The training details are missing.
2. Technical contribution is weak. I do not see significant improvement with previous multi-modal methods (especially the core design by aligning latent space).
3. The experimental design is strange. The title of the paper is point cloud understanding. I think the authors should provide results for point cloud understanding tasks (e.g., point cloud classification, at least). It’s not clear why the authors only perform experiments on point cloud generation.

**Questions:**

Please refer to weaknesses.

---

### Official Review · Reviewer_AdrN · 2023-11-01

**Soundness:** 1 poor
**Presentation:** 1 poor
**Contribution:** 1 poor
**Rating:** 3
**Confidence:** 4

**Summary:**

This paper introduces UniPoint-LLM, a 3D multi-modal large language model that combines both comprehension and generation capabilities. The authors employ the llama-adapter as the model backbone, utilize the shape-e as the point cloud decoder, and supervised fine-tuning on the Cap3D dataset for achieving 3D Visual Question Answering and text-conditioned 3D generation.  By introducing a Multimodal Unified Feature Space (MUTS), UniPoint-LLM achieves multi-modal understanding of images and point clouds.

**Strengths:**

This paper is the first to try to explore a relevant problem of unified comprehension and generation in the 3D domain.

**Weaknesses:**

1. This article appears to be incomplete, as I can't locate Table 2: Zero-shot 3D object classification.
2. The authors mention that UniPoint-LLM has the capability to perform 3D VQA tasks, but there is a lack of corresponding experiments or performance demonstrations.
3. All the modules, including llama-adapter and shape-e, have already been proposed. And the training dataset is from Cap3D. This paper appears to be incremental in nature.
4. I suggest enhancing the efficacy demonstration and quantitative assessment of 3D VQA like PointLLM [Wang et al., 2023] and exploring text-conditional 3D point cloud editing like instructP2P [Xu et al., 2023] and VPP [Qi et al., 2023].

[Xu et al., 2023] Pointllm: Empowering large language models to understand point clouds. In arXiv, 2023.

[Xu et al., 2023] InstructP2P: Learning to Edit 3D Point Clouds with Text Instructions. In arXiv, 2023.

[Qi et al., 2023] VPP: Efficient Conditional 3D Generation via Voxel-Point Progressive Representation. In NeurIPS, 2023.

**Questions:**

See Weaknesses.